# Impact of a Topical Anaesthesia Wound Management Formulation on Pain, Inflammation and Reduction of Secondary Infections after Tail Docking in Lambs

**DOI:** 10.3390/ani10081255

**Published:** 2020-07-24

**Authors:** Luis Miguel Ferrer, Delia Lacasta, Aurora Ortín, Juan José Ramos, María Teresa Tejedor, Marta Borobia, María Pérez, Enrique Castells, Marta Ruiz de Arcaute, Héctor Ruiz, Peter Andrew Windsor

**Affiliations:** 1Animal Pathology Department, Instituto Agroalimentario de Aragón-IA2, Veterinary Faculty of Zaragoza, Universidad de Zaragoza-CITA, C/Miguel Servet 177, 50013 Zaragoza, Spain; lmferrer@unizar.es (L.M.F.); aortin@unizar.es (A.O.); jjramos@unizar.es (J.J.R.); mborobia@unizar.es (M.B.); mariaperezb3@hotmail.com (M.P.); martarda@unizar.es (M.R.d.A.); hectorruiz353@gmail.com (H.R.); 2Anatomy, Embryology and Animal Genetics Department, Veterinary Faculty of Zaragoza, CIBER CV, Universidad de Zaragoza-IIS, C/Miguel Servet 177, 50013 Zaragoza, Spain; ttejedor@unizar.es; 3Centro Clínico Veterinario, C/Madre Genoveva Torres Morales 8, 50006 Zaragoza, Spain; kikevet39@gmail.com; 4Sydney School of Veterinary Science, The University of Sydney, Camden, NSW 2570, Australia; peter.windsor@sydney.edu.au

**Keywords:** lambs, tail docking, pain response, wound infection, cortisol, serum amyloid A

## Abstract

**Simple Summary:**

Tail docking is routinely conducted in sheep husbandry around the world, often without anaesthesia or analgesia despite recognition that it is a painful process. Several methods are used, although the Council of Europe states that surgical methods with anaesthesia should be used to dock the tail, with these recommendations copied into national legislation by most European countries. However, no studies have been documented comparing the use of general anaesthesia and stitching. This study compared different surgical approaches with and without general anaesthesia, and with and without surgical stitching, plus examined the efficacy of applying a commercially available wound gel spray containing topical anaesthetics and an antiseptic, immediately on tail amputation. Results demonstrated that lambs display more pain when the surgery included stitching and these wounds became more readily infected, potentially requiring antibiotic therapy. As the generation of potential antimicrobial resistance and drug residues issues in food-producing animals from over-use of antibiotics is an emerging one health consideration, the availability of a topical anaesthetic formulation that provides pain relief and contains an antiseptic that reduces secondary infections and hasten wound healing, is of potential importance for improving husbandry procedures such as tail docking.

**Abstract:**

We examined several procedures for surgical tail docking; with and without general anaesthesia (GA), including the use of a topical wound gel formulation to provide pain relief (PR) and improve healing after surgery, containing local anaesthetics lignocaine and bupivacaine, with cetrimide and adrenalin. Forty-four lambs were recruited into four equal cohorts: Groups A and C, the tail was excised with a scalpel without anaesthesia or stitches; Groups B and D, the tail was surgically excised and stitched under GA; Groups C and D wounds were immediately sprayed with PR. Behavioural observations identified that Groups A and C displayed significantly less pain-related behaviours than Groups B and D shortly after the procedure, especially if treated with PR. Similarly, the mean of days when animals showed no signs of wound infection was longer in the groups not undergoing stitching. Finally, treatment with PR appeared to reduce the cortisol response and avoided the elevation of serum amyloid A in lambs where the tail was excised without general anaesthesia. In conclusion, surgical tail-docking without GA but where wounds are immediately sprayed with PR, appears as an affordable and more welfare-appropriate method for conducting tail docking in lambs.

## 1. Introduction

Amputation of the tail in livestock or tail docking, is a routine husbandry procedure in sheep production worldwide for preventative management of myiasis [1,2,3] and improved fertility [4]. Although numerous studies have demonstrated that tail docking causes acute pain in lambs [5,6,7], it remains a common procedure, often performed without pain relief. Despite justification for such aversive husbandry procedures to manage animal health and production issues, there is a growing concern for the welfare of animals undergoing these procedures, with demands they cease or at least include pain relief [8].

Tail docking practices vary between regions and countries, most commonly involving either rapid amputation by surgical excision, or delayed ischaemic removal by application of a rubber ring, despite clear evidence that it causes extreme pain. In the UK, it was estimated that 90% of farmers remove tails, with use of rings most common (86%), followed by surgical (3%) and other methods (2%) [7,9]. The acute ischaemia in the tail distal to the rubber ring leads to localised tissue necrosis that, after an extended period, eventually causes sloughing. Tail docking by surgery involves severing the tail using a sharp knife or scalpel, or a hot knife or docking iron to cauterise the wound [10,11]. These procedures have mostly been performed by farmers or their contractors without analgesia or anaesthesia as the use of anaesthesia usually requires costly participation or supervision of a veterinarian and until recently, there have been no pain relief products available for use by farmers [8,10]. The Council of Europe (CoE), in 1992 [12], stated in Article 30 that mutilations are not permitted, except for tail docking (as long as sufficient length of the tail is left to cover anus and vulva). It is said that rubber rings should be avoided, and surgical methods with anaesthesia should be used. The CoE recommendations are copied into national legislation by most European countries, and there are only three countries that ban lamb docking in Europe: Lithuania, Finland, and Sweden. With the growing demand for humane management of production animals, there is a societal expectation that any suffering from such aversive procedures is minimised. There is an urgent need for practical and affordable approaches to tail docking with pain relief suitable for use by farmers, that reduces animal suffering.

As the evaluation of pain responses is difficult, especially in lambs, mainly since behaviours that reflect pain are not displayed as intensively as in other species [13], behavioural and physiological methods may be used to evaluate post-procedural responses following tail docking. Global ratings of pain have been used to assess animal pain in clinical situations [14,15], with several rating methods developed to directly assess pain-related behaviour [16,17]. The activity of the hypothalamic-pituitary-adrenocortical (HPA) system has been extensively measured to indirectly gauge the distress of tail docking and the effectiveness of analgesics [18,19,20]. Plasma or serum cortisol concentrations provide an indication of HPA activity and are useful indices as, within certain limits, secretion occurs in a graded way in response that may reflect the presumed noxiousness of different experiences. The concentration of acute phase proteins (APPs) changes in response to inflammation, infection, trauma, and stress [21,22]. Although knowledge of the acute phase response is less in small ruminants than in other species, serum amyloid A (SAA) and haptoglobin (HP) are considered the main APPs in these species. They can increase up to 10–100 times in response to stimulation [23], and elevated concentrations in small ruminants have been reported under certain infectious conditions, experimental inflammatory stimulus [23,24], and in response to transportation stress [25]. The SAA response is generally stronger than that of HP, and its concentration tends to increases rapidly, while HP increases at a slower rate [21]. The study of the evolution of SAA levels following tail docking could enable measurement of the tissue trauma and the recovery of the animals following the different procedures used.

In this study, two different procedures of surgical tail docking, with and without general anaesthesia (GA) and with and without surgical stitching, were assessed. In addition, the immediate application of a topical ‘spray-on’ wound management formulation after both techniques was assessed. This therapy contains the local anaesthetics lignocaine and bupivacaine, plus cetrimide and adrenalin, in a gel matrix (Tri-Solfen^®^, Bayer Animal Health, Gordon, NSW, Australia), that was developed in Australia to provide pain relief (PR) during aversive husbandry procedures. The rapid uptake and widespread adoption of this product in Australian livestock husbandry use has been described as a ‘pain management revolution’, with increased traction in other jurisdictions leading to studies for use in managing wounds in humans [26]. The current study involved assessment of pain-related behaviour by a numerical rating scale (NRS), developed from a combination of previous field observations and reports of behavioural changes in response to mulesing and/or castration [17]. The rate of wound healing and the onset of secondary bacterial infection was assessed. Further, the study included measurement of the concentration of serum cortisol (SC), and the acute phase protein serum amyloid A.

## 2. Materials and Methods

All procedures were carried out under Project Licence PI 37/19 approved by the Ethics Committee for Animal Experiments from the University of Zaragoza. The care and use of animals were performed accordingly with the Spanish Policy for Animal Protection RD53/2013, which meets the European Union Directive 2010/63 on the protection of animals used for experimental and other scientific purposes.

### 2.1. Selection of Farm, Animals, and Treatment Groups

The selected farm was located near Zaragoza (Spain) and follows the prototype of the production system in the area. This involves a semi-intensive system for ternasco-type lamb production, where sheep are housed at the end of gestation and during lactation, with lambs kept indoors until slaughter at approximately three months of age. Forty-four 25 to 35-day-old Rasa Aragonesa female lambs with similar weights (8–10 kg) were recruited for this study and randomly divided into four equal cohorts (*n* = 11). The four equal cohorts of lambs were: Group A, the tail was excised with a scalpel without anaesthesia or stitches; Group B, the tail was surgically excised and stitched under GA; Group C, the tail was removed with a scalpel without prior anaesthesia and treated immediately with PR; Group D, the tail was surgically excised and stitched under GA and treated with PR applied before suturing of peri-wound skin to close the wound. A single application of the pain relief product Tri-Solfen^®^ was sprayed on the wounds, using a product applicator (gun) providing a total dose of 1.5 mL per animal up to 10 kg. No additional treatments, including antibiotics, were applied to the lambs throughout the entire study. All surgery was performed on the same day and following the procedure, the animals were reintroduced into their production lot, each consisting of 96 sheep and 150 lambs. The facilities were cleaned and disinfected before each lambing period and bedding straw was used.

### 2.2. Tail Docking Surgical Procedures

For surgery under general anaesthesia in Groups B and D (*n* = 22), dexmedetomidine was used as pre-anaesthetic at a dose of 2.5–5 mg/kg of body weight intravenously (iv), and buprenorphine at a dose of 0.01 mg/Kg iv, inoculating propofol at a dose of 1 mg/Kg iv when the animal showed signs of sedation. This regimen provided between 5 to 10 min of anaesthetic to conduct the surgical amputation, performed, after disinfecting the skin with iodine solution, by incising with a scalpel between the 68th coccygeal vertebrae, then suturing of peri-wound skin to close the wounds with loose stitches. In lambs belonging to Group D, PR was sprayed on the wounds with a spray gun that applied 1.5 mL in each application, just prior to suturing. The time required to perform the entire procedure was recorded.

For surgery without general anaesthesia, in Groups A and C (*n* = 22), the surgical amputation of the tail was performed using the traditional method, consisting of producing haemostasis with the pressure of the fingers proximal to the sixth coccygeal vertebrae, then docking the tail distal to that level with a scalpel. Finger pressure was maintained until there was nil or only minimal haemorrhage, the wounds were not sutured, plus PR was sprayed on the wounds of Group C with a spray gun that applied 1.5 mL in each application. The traditional procedure was conducted by the farmer who usually did it on the farm, with an additional person applying the topical PR treatment on the lambs that belonged to Group C. The lambs were restrained by the farmer between his legs during the intervention. The time required to perform the whole procedure was recorded.

### 2.3. Assessment of Pain-Related Behaviour

Pain-related behaviour after tail docking was assessed using the NRS developed previously [8]. A trained scientist blinded to treatment with PR, observed the lambs within 5 min following their release into the lot with their mothers, when they could freely express their natural behaviour (R1), 2.5 h (R2) and 5 h (R3) after tail docking. For lambs recovering from GA release occurred within 10 min of completion of the surgery, whereas for lambs without GA, release occurred within a minute following the surgery. Individual lambs were ascribed an NRS score between 0 and 3, where: 0 = no pain-related behaviour; 1 = mildly abnormal posture, gait or behaviour, including mild kyphosis without hyperextension of hindlegs, ventral recumbency with hindlegs partially extended or mild stiffening of gait without overt limping or leg dragging; 2 = moderately abnormal posture, gait or behaviour, including ‘statue standing’ with head down and prominent kyphosis, moderate stiffening or slowing of gait or hyperextension and/or abduction of hindlegs, ventral recumbency with hindlegs fully extended; 3 = displaying severely abnormal posture, gait or behaviour, including marked agitation with twisting or writhing, high frequency of postural change from lying to kneeling or standing, distressed vocalisation, lateral or prostrate lying, kneeling, dog sitting or tremors, shaking or lip curling. All the procedures and the subsequent behaviours of the lambs were recorded.

### 2.4. Wound Management

Following the tail docking procedure, the animals were examined daily for 23 days, with lesions photographed. This enabled analysis of wound healing after tail docking and the detection of secondary infections. Wound management was evaluated as days in which the animals showed no signs of wound infection. It was considered that a wound was infected when yellow or greenish-coloured surface or draining exudate was observed.

### 2.5. Haematology

Samples of 3 mL of whole blood were taken from the jugular vein through a vacutainer system to perform haematology in all the animals prior to tail docking (T0) and seven days later (T1). Haematology was performed using an automatic haematological counter Vet-ABC (DIVASA-FARMAVIC S.A., Barcelona, Spain). Measured parameters were leucocytes (10^9^ cells/L), erythrocytes (10^12^ cells/L), haemoglobin (g/L), PCV (packed cell volume; %), platelets (10^9^ cells/L), MCV (Mean Corpuscular Volume; fl), MCH (mean corpuscular haemoglobin; pg) and MCHC (mean corpuscular haemoglobin concentration; g/L).

### 2.6. Serum Cortisol Concentration (SC)

For assessment of SC following the different tail docking procedures, 3 mL blood samples without anticoagulants from the jugular vein were collected from each lamb into a vacutainer without anticoagulants, prior to tail docking (C0), and at 30 min (C1), 5 h (C2) and 48 h (C3) intervals post-tail excision. Sera were removed and stored at −20 °C until SC concentration (ng/mL) was analysed using a competitive ELISA assay (Salivary Cortisol ELISA SLV-2930, DRG Diagnostics, Marburg, Germany). The intra- and inter-assay CV were respectively 2.6% and 4.3% and the sensitivity was 0.09 ng/mL.

### 2.7. Serum Amyloid A Concentration (SAA)

In the samples that coincided in time, the same extracted serum was used to carry out the quantification of SC and SAA. For SAA, the 3 mL blood samples without anticoagulants were designated as: prior to tail docking (S0); and at 5 h (S1); 48 h (S2) intervals; plus, seven days later (S3). Sera were obtained and stored at −20 °C until analysed to determine SAA concentrations (ng/mL), using a solid-phase sandwich ELISA kit (PHASE TM Serum Amyloid A Assay, Tridelta Development Ltd., Maynooth, Ireland). The intra- and inter-assay CV were respectively, 5.0% and 11.4% and the sensitivity was 0.3 µg/mL.

### 2.8. Statistical Analysis of Results

Statistical analyses were performed using IBM SPSS statistics version 26 (2019) software (IBM, Armonk, NY, USA). Shapiro-Wilk’s test was used to assess normality of considered variables. Leucocyte count and pain-related behaviour were analysed using a non-parametric test (the Mann–Whitney U test and Wilcoxon test, where necessary). The proportion of individuals with leucocytosis were compared among groups by the chi-squared test. A logarithmic transformation (log^10^) was used to normalise SC and SAA concentrations before statistical analysis. For each procedure (with or without the use of general anaesthesia), a two-way mixed ANOVA was run to understand the effects of treatment (with or without PR) and time on erythrocyte count, log^10^ SC concentration (C0 to C3), and log^10^ SAA concentration (S1 to S4). Levene’s test and Box’s M test were applied to assess the homogeneity of variances and covariances, respectively. When adequate, Mauchly’s test was applied to test the assumption of sphericity for two-way interaction. Finally, wound management data were analysed using proportion comparison by Fisher’s exact test and survival analysis (the Kaplan–Meyer method and Breslow test), evaluating the days until the wound was found infected. The possible association between the level of SAA at the different sampling times and the wound healing was investigated by determining the Pearson correlation. In every test, *p* values < 0.05 were considered statistically significant. When significant differences among more than two groups were found, pairwise comparison with Bonferroni correction was carried out.

## 3. Results

The time required to perform the surgery under general anaesthesia and subsequent stitching of each lamb exceeded 10 min and the traditional method without GA took less than a minute. The estimated cost of anaesthetic and the material necessary for the surgery was 7.90 euros/lamb and the economic cost in material in the traditional method was 0.50 euros/lamb. The cost of PR per animal treated was 0.50 euros. No lamb mortality was recorded throughout the study.

### 3.1. Assessment of Pain-Related Behaviour

As displayed in Table 1 and Figure 1, the groups tail docked by the traditional method (A and C) displayed significantly less pain (*p* = 0.004) than the others (B and D) shortly after (R1) the procedure (1.27 and 0.68 vs. 2.36 and 2.04) (Table 1). In addition, lambs treated with PR in the group tail-docked without general anaesthesia (C) displayed a lower pain score than those untreated (A) (0.68 vs. 1.27) (*p* = 0.013).

By 2.5 h after tail docking (R2), the NRS scores tended to decline and equalize. In groups without PR (A and B), the mean of the pain response was 1.14 and 1.09, whereas, in groups treated with PR (C and D), the mean was 0.86 and 0.81, respectively, although these differences were not found significant (*p* > 0.05). Finally, at 5 h (R3) most of the animals, independent of the group they belonged to, were no longer displaying pain behaviours (Table 1).

### 3.2. Wound Management

In the period studied (23 days), 73% (16/22) of lambs in the two groups not undergoing surgical stitching (A and C), and 91% (20/22) of lambs in the two groups undergoing surgical stitching (B and D), developed signs of wound infection, although these proportions did not differ significantly (*p* = 0.240). The mean of days in which the animals showed no signs of wound infection after tail docking was higher in traditional groups (Table 2; A: 7.54 and C: 8.73 vs. B: 4.27 and D: 3.64). Significant differences were found when comparing the two methods, with and without stitching among those treated with PR (C vs. D; *p* = 0.015), and it was observed that group C, being those lambs whose tail was excised without stitching and the wounds were treated with PR, had the higher number of days without signs of wound infection.

### 3.3. Haematology

For leucocyte counts, no significant differences were found at T0 and T1 between groups. In group B, only the leucocyte count significantly increased in the second sampling (T1), seven days after the procedure (*p* = 0.041) (Table 3). Although no significant differences were found (*p* = 0.553), the number of lambs with leukocytosis in the second sampling was higher in groups not treated with PR, A and B (A: 36.4% and B: 45.5% vs. C: 18.2% and D: 27.3%). For the erythrocyte count, the values were significantly higher in T0 than in T1 in all the cases (*p* < 0.01) and the erythrocyte count was lower when tail docking was carried out without anaesthesia (*p* = 0.009) (Table 3). No differences were found in any of the other haematological parameters examined.

### 3.4. Serum Cortisol Concentration (SC)

SC results are displayed (Figure 2 and Appendix A). In groups tail-docked under general anaesthesia (B and D), SC concentration did not differ significantly between the group treated (D) and not treated with PR (B) (*p* = 0.919), and SC concentration did not change significantly over time (*p* = 0.205). In groups tail-docked without general anaesthesia (A and C), although no significant differences were found for SC concentration between them (*p* = 0.162), in these two groups, SC peaked at 30 min post-tail removal (C1), and decreased thereafter, with C1 values significantly higher than the remaining values (*p* < 0.01). Although statistically significant differences were not detected between the groups (*p* = 0.162), SC at C1 and C2 for Group C were lower than for Group A (Figure 2).

### 3.5. Serum Amyloid A Concentration (SAA)

SAA results are displayed (Figure 3 and Appendix A). In groups tail-docked under general anaesthesia (B and D), SAA increased from S0 to S2 and decreased in S3, with the S2 value significantly higher than the rest of the values (*p* < 0.01). The group tail-docked with PR (D) had consistently lower SAA levels than the group without this wound treatment (B), although statistically significant differences were not determined between these groups (*p* = 0.604). In groups tail-docked without general anaesthesia (A and C), SAA evolved similarly as a function of time, with only the S2 value differing significantly from S3 (*p* = 0.003). A detailed examination of data from Groups A and C found that SAA concentration in Group A clearly showed a maximum in S2. The SAA in the PR treated group (C) failed to achieve the maximum concentrations at S2 and were similar to S0 and S1, decreasing by S3, although statistically significant differences were not detected between lambs treated and non-treated with PR (*p* = 0.913).

Finally, a significant bilateral (0.014) negative correlation (−0.368) was found between the days until the wound was found infected and the SAA concentration at S2 (48 h post-tail docking), the time at which significant elevations of SAA are expected after a noxious stimulus.

## 4. Discussion

Tail docking is routinely conducted in sheep husbandry around the world, often without anaesthesia or analgesia despite recognition that it is a painful process [5,6,7]. Several methods can be used to perform tail docking, most commonly the use of a sharp knife, scalpel, hot knife or rubber rings [11]. All these techniques have been demonstrated to be painful for lambs. The Council of Europe states that surgical methods with anaesthesia should be used to dock the tail [12]. To the authors’ knowledge, no studies have been documented comparing the use of general and local anaesthesia in tail docking in lambs. The present study was developed to compare two surgical approaches applied to tail docking and examine if the use of a commercially available ‘spray-on’ wound formulation containing two topical anaesthetics (PR) can help alleviate pain and assist recovery following tail amputation. The study compared surgical excision of the tail under general anaesthesia and subsequent stitching, with the traditional method, where the tail is excised with a scalpel without anaesthesia or stitching.

The results demonstrated that the lambs displayed more pain when the surgery included stitching and the wounds become more readily infected. This approach also involves costs in time and money that do not appear to be compensated by improved animal welfare. The time required to perform the surgery with the stitching of each lamb exceeded 10 min, with the lambs having to be managed twice; once to carry out the sedation and later the surgery. This compared to the few seconds required to conduct the traditional method. Further, if the procedure is performed in very young lambs and the effects of anaesthesia are prolonged, maternal-filial recognition (referred to as ‘mothering up’ in some countries) may be compromised. In our study, it was observed that the animals tail docked under general anaesthesia suffered significantly more pain during recovery from anaesthesia than those that did not have GA. This likely reflects that surgery with stitching inflicts several painful procedures, including: injection of the intravenous pre-anaesthetic; injection of the intravenous anaesthetic; the tail docking; and, finally, the application of stitches following excision that appears to increase pain during recovery. Further, the increased risk of death of lambs under general anaesthesia is a consideration, as is the generation of potential drug residues in food-producing animals. In companion animals, general anaesthesia is done under much safer conditions than in production animals, and a mortality rate of between 0.1 to 0.24% is reported [27]. In horses attended in a veterinary hospital, the mortality rate following general anaesthesia increased to 1.4% [28]. Although no mortalities were observed in our study, the percentage following general anaesthesia in lambs under farm conditions is likely to exceed those in companion animals and horses.

It is considered critically important for the sustainability of livestock production that improved animal welfare occurs on farms. The identification of efficacious and economically affordable protocols for use by farmers conducting aversive livestock husbandry procedures is increasingly necessary. These protocols should be safe and sufficiently mitigate post-procedural pain after tail docking to ensure that the improved animal welfare is clearly visible to farmers, motivating their continued investment in products used, preferably avoiding the costly need for a veterinarian to be present. In our study, we found the farmer-applied, spray-on topical anaesthetic formulation (Tri-Solfen^®^) successfully mitigated the pain displayed by lambs after tail docking. This product was designed in Australia for wound pain management in sheep, providing maximum adhesion and duration of effect, acting directly on exposed peripheral nervous tissue, eliciting a reversible block of the conduction of the nociceptive signals responsible for the sensation of pain. The product has been shown previously to be effective for tail docking with a hot knife [10]. When applied to open wounds, PR effects rapid and profound local anaesthesia, with our results in agreement with others that have tested the use of topical anaesthetics to mitigate pain in lambs after traumatic procedures [8,10,17,18,29]. Although the immediate use of topical anaesthesia after a traumatic practice could seem contradictory, postoperative pain may be more effectively addressed as the actives are delivered directly to the traumatised nerve fibres and tissues that are the source of pain. In addition, the application of topical anaesthesia prevents the subsequent pain escalation response pathway, consequently improving the general welfare of lambs [10].

As it can be difficult to quantify pain in lambs [13], most studies report a combination of approaches. We used several methods to assess both behavioural and physiological changes related to pain. The chosen method to assess the pain-related behaviour used a numerical rating scale (NRS) that was developed in a previous study [17] and has been used in several subsequent studies [10,29]. The most accurate pain-related behaviour measurements are obtained with data collected from the first 60 min after treatment [16], although as our study used a measure at 30 min, acceptable accuracy may also be obtained with shorter periods. Further, behaviours recorded as events rather than time spent carrying out the activity, are considered as preferred indicators of pain, due to the broad variation in response to the same stressor [30]. In our study, combining different activities into a single index compensates to some extent for this individual variation [16].

An additional objective of the present work was to evaluate the progression of tail docking wounds following either of the two procedures applied, assessing impacts of using the topical anaesthetic wound spray PR. Almost 82% of the studied lambs developed signs of wound infection after tail docking. This high percentage of infections likely reflects that the lambs remained stabled throughout the period of study, and antibiotic treatment was not applied during the procedure, despite this being a usual practice. However, the mean of days on which the animals showed no signs of wound infection were higher in the two groups not undergoing surgical stitching. This suggests that closing the wound with the surgical stitches and not treating the animal with antibiotics in a potentially contaminated environment, increases infection risk, delaying wound healing. Although the use of antibiotics is a common practice after surgical procedures in animals, this increases both the cost of the procedure and risks of antimicrobial residue and resistance. Further, the practice is in violation of the antibiotic use reduction policy as promulgated by the EU. Importantly, it was observed that best results were obtained in the group of lambs where the tail was excised without stitching, and the wounds were treated with PR. This accords with reports that PR improves wound healing in aversive husbandry procedures conducted in Australia, including mulesing [17] and tail docking [10] of lambs and castration of calves [31]. Of note is that these studies were conducted in extensively-raised animals, and there was no evidence of delayed wound healing due to high rates of infection.

Plasma or serum cortisol concentration have been widely used to assess the responses to painful experiences in lambs [18,19,20]. However, there is some controversy on the application of cortisol in studies of pain, considered as an indirect measure of stress, not pain. The handling and short-term removal of the lamb from the dam to carry out tail docking is an important source of stress to the lamb [19], and stressors such as restraint or social isolation increase cortisol levels in sheep [31]. Further, as serum cortisol shows a ‘ceiling effect’ this limits its use for grading severely painful procedures [16], plus cortisol will be elevated if there is haemorrhage present, limiting the comparison of surgical versus non-surgical procedures [32].

In our study, in groups tail-docked without general anaesthesia, serum cortisol concentrations peaked at 30 min post-tail removal, decreasing thereafter, as widely reported in previous studies [19,33]. However, in groups subjected to general anaesthesia, this cortisol peak was absent, despite these lambs displaying significantly increased pain-related behaviours following tail docking. This was most likely due to the fact that the animals, when anesthetized, did not suffer the stress of the procedure, but the consequent pain that is displayed, is not reflected in the cortisol peak. This observation corroborates other evidence, particularly in calves, that serum cortisol is measuring stressful management, such as handling, restraint and potentially haemorrhage, rather than primarily the pain after tail docking [30,31]. The absence of effect of NSAIDs on the acute cortisol response has been observed in other studies [18,34], a further indication that serum cortisol responses are measures of stressful experiences, rather than just pain. In our study, the PR treated group without general anaesthesia and subsequent stitching had SC concentrations lower than in the PR untreated group, although these differences were not statistically significant. These results could imply that treatment with PR did reduce the cortisol response elicited after the surgery without general anaesthesia, although the low statistical power due to small sample sizes used, could well have prevented these differences as being significant. This is in accordance with previous findings in which topical anaesthetic applied to the castration wound reduced the cortisol peak at 30 min [18], and that small group sizes are likely to compromise statistical significance of trends in lower cortisol responses between treated and untreated groups of castrated [31] and dehorned calves [32].

The quantification of the acute phase proteins can assist the assessment of infection, inflammation, trauma, tissue damage and stress in animals, and are applied in order to improve the detection, diagnosis, prognosis and monitoring different diseases and pathological processes of domestic animals [21,22,24]. Elevated concentrations of HP and SAA have been reported in small ruminants under certain infectious conditions, experimental inflammatory stimuli [23,24], and in response to transportation stress [25]. However, although HP is an effective marker for monitoring processes such as tail docking and castration in cattle [21,35], HP levels remain unchanged in lambs following these processes [35]. In our study, a significant increase in SAA concentration was observed in the groups of lambs submitted to surgery with stitching following tail docking. The SAA response in lambs is generally stronger than that of HP, and its concentration tends to increase more rapidly, reaching maximum concentrations within 24 to 48 h after stimulation, with a rapid normalisation within four to seven days, if no further stimulus occurs [21]. Our results are in accordance with this finding since the highest concentration of SAA in this study was measured at 48 h post-tail docking. A rise in SAA concentration at this time point, although not significant, was also observed in the group tail-docked without general anaesthesia and not treated with PR, whereas, in the PR treated group, SAA concentration remained unchanged. Although elevated concentrations of APPs also occur with some conditions not generally associated with inflammation, infection or tissue damage, such as stress, the effect of stress on serum concentration of APPs remains controversial since it is difficult to distinguish the effect of stress from the effect of trauma [21]. In our study, the increase in SAA concentration observed is likely a consequence of pain associated with the procedure and a reflection of the tissue damage, inflammation and particularly, wound infection. Groups tail-docked with general anaesthesia, in which SAA rise was significant, showed significantly higher pain-related behaviour after tail docking than the others, with the surgical stitching in these lambs likely to be responsible for greater tissue damage, inflammation and risk of infection. Of interest, the group not undergoing surgical stitching that was treated with PR suffered the least pain and showed unchanged SAA concentration at 48 h post-tail docking. Further, stress does not seem to be involved in the acute phase response observed, as in groups submitted to surgery with stitching, serum cortisol concentrations did not peak at 30 min post-tail removing.

The relationship between wound healing and SAA in our study deserves consideration, as it is likely that wound infection could have contributed to the elevation in SAA at 48 h post-tail docking. The mean of days on which the animals showed no signs of wound infection was significantly shorter in the two groups undergoing surgical stitching, where a significant increase in SAA concentration was also measured. Further, in the group of lambs where the tail was excised without stitching, and the wounds were treated with PR, the SAA concentration did not change within 48 h following the procedure, and these animals develop lower signs of wound infection. A significant bilateral negative correlation was found between the days until the wound was found infected, and the SAA concentration at 48 h post-tail docking. Although the intensity of association between both parameters is medium, these results appear to indicate that higher SAA concentration 48 h after tail docking could detect a worse evolution of the wound repair process, due to more rapid onset of infection. These promising results indicate that SAA could be a useful biomarker for monitoring acute phase responses after tail docking in lambs and predict the evolution of wound healing, although further research is needed to corroborate these findings.

This study suggests that the method traditionally performed by farmers appears to demonstrate clear advantages compared to tail docking under general anaesthesia. Importantly, the estimated cost of PR per animal treated in this study was Euro 0.50, compared to Euro 7.90 for the GA protocol.

## 5. Conclusions

The results from this study and other reports are encouraging, suggesting that surgical tail docking without general anaesthesia and subsequent stitching but where wounds are immediately sprayed with Tri-Solfen^®^ is an affordable and efficacious method of conducting the procedure by farmers that improves animal welfare. This topical anaesthesia wound treatment formulation alleviates pain and hastens healing, reducing secondary infections and avoiding the unnecessary use of antibiotics that may contribute to the risks of developing antimicrobial resistance and residues.

## Figures and Tables

**Figure 1 animals-10-01255-f001:**
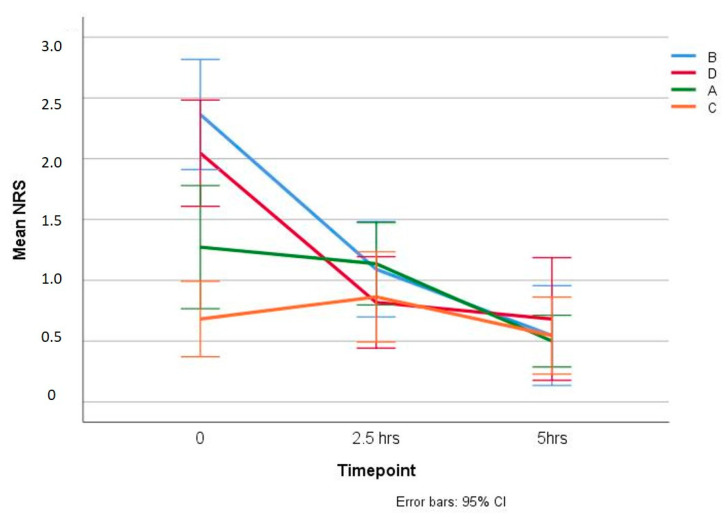
Mean of pain-related behaviour score (NRS) in the four Groups A, B, C, and D, shortly (R1), 2.5 h (R2) and 5 h (R3) following tail docking (Error bar: 95% CI). Groups A and C, the tail was excised with a scalpel without anaesthesia or stitches; Groups B and D, the tail was surgically excised and stitched under GA; with Groups C and D immediately sprayed with PR.

**Figure 2 animals-10-01255-f002:**
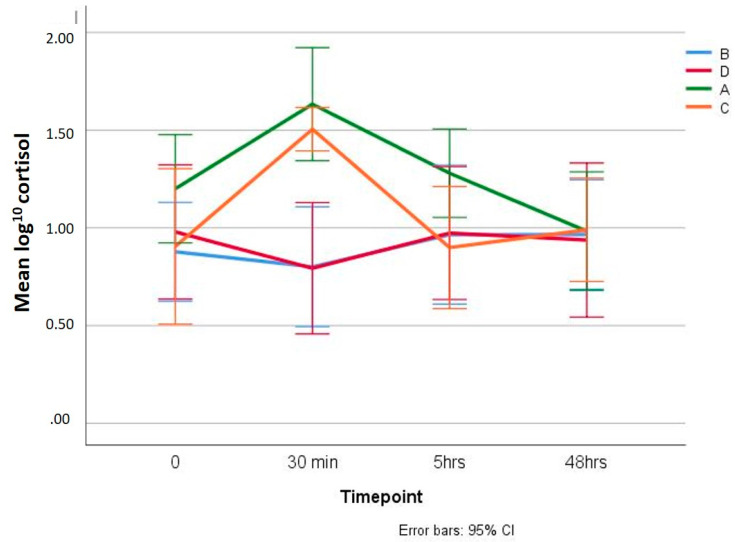
Mean of log^10^ cortisol in the four Groups A, B, C, and D, prior to tail docking (C0), and at 30 min (C1), 5 h (C2) and 48 h (C3) intervals post-tail excision (Error bar: 95% CI). Groups A and C, the tail was excised with a scalpel without anaesthesia or stitches; Groups B and D, the tail was surgically excised and stitched under GA; with Groups C and D immediately sprayed with PR.

**Figure 3 animals-10-01255-f003:**
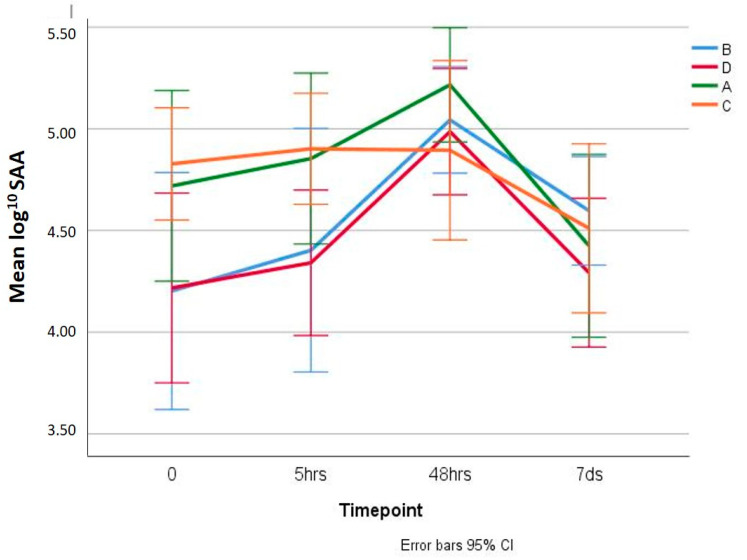
Mean of log^10^ SAA in the four Groups A, B, C, and D, prior to tail docking (S0), and at 5 h (S1) and 48 h (S2) intervals, plus seven days later (S3) (Error bar: 95% CI). Groups A and C, the tail was excised with a scalpel without anaesthesia or stitches; Groups B and D, the tail was surgically excised and stitched under GA; with Groups C and D immediately sprayed with PR.

**Table 1 animals-10-01255-t001:** Mean and SD of pain-related behaviour score (NRS) following tail docking by “Surgery” (conducted under general anaesthesia (GA) with stitching; Groups B and D) or “Traditional” (conducted without general anaesthesia or stitching; Groups A and C), with or without treatment with pain relief (PR) (yes or no, respectively). Only significant differences were found for R1: ^a,b^: different letters mean significant differences between groups within procedure (*p* < 0.05), ^A,B^: different letters mean significant differences between procedures (*p* < 0.05).

Time	Procedure	Pain Relief Treatment (Group)	N	Mean ± SD
R1	Traditional	No (Group A)	11	1.27 ± 0.754 ^a,A^
		Yes (Group C)	11	0.68 ± 0.462 ^b,A^
	Surgery	No (Group B)	11	2.36 ± 0.674 ^a,B^
		Yes (Group D)	11	2.04 ± 0.650 ^a,B^
R2	Traditional	No (Group A)	11	1.14 ± 0.504
		Yes (Group C)	11	0.86 ± 0.552
	Surgery	No (Group B)	11	1.09 ± 0.584
		Yes (Group D)	11	0.82 ± 0.560
R3	Traditional	No (Group A)	11	0.50 ± 0.316
		Yes (Group C)	11	0.54 ± 0.472
	Surgery	No (Group B)	11	0.54 ± 0.610
		Yes (Group D)	11	0.68 ± 0.751

**Table 2 animals-10-01255-t002:** Mean and SE of days free of signs of infection, following tail docking by “Surgery” (conducted under general anaesthesia with stitching; Groups B and D) or “Traditional” (conducted without general anaesthesia or stitching; Groups A and C), with or without treatment with PR (yes or no, respectively). ^A,B^: different letter mean significant differences between procedures (*p* < 0.05).

Procedure	Pain Relief Treatment (Group)	Mean ± SE
Traditional	No (Group A)	7.54 ± 2.867 ^A^
	Yes (Group C)	8.73 ± 2.666 ^A^
Surgery	No (Group B)	4.27 ± 1.805 ^B^
	Yes (Group D)	3.64 ± 1.869 ^B^

**Table 3 animals-10-01255-t003:** Mean and SD of the leucocyte (WBC, 10^9^ cells/L) and erythrocyte counts (RBC, 10^12^ cells/L) at T0 (immediately prior to tail docking) and T1 (seven days after procedure), where “Surgery” was tail docking performed under general anaesthesia (Groups B and D), “Traditional” was tail docking conducted without general anaesthesia (Groups A and C), and ‘yes’ or ‘no’ was with or without PR, treatment applied, respectively. ^a,b^; different letters mean significant differences between T0 and T1 (*p* < 0.05). ^A,B^; different letters mean significant differences between Traditional and Surgery groups for RBC in both T0 and T1 (*p* < 0.05).

Variable	Group	Pain Relief Treatment (Group)	N	T0	T1
				Mean ± SD	Mean ± SD
Leucocyte count(WBC, 10^9^ cells/L)	Traditional	No (Group A)	11	9.85 ± 2.799 ^a^	10.40 ± 2.999 ^a^
		Yes (Group C)	11	10.14 ± 5.358 ^a^	11.18 ± 8.253 ^a^
	Surgery	No (Group B)	11	10.27 ± 3.604 ^a^	12.34 ± 3.803 ^b^
		Yes (Group D)	11	8.80 ± 3.133 ^a^	10.34 ± 2.202 ^a^
Erythrocyte count(RBC, 10^12^ cells/L)	Traditional	No (Group A)	11	10.78 ± 1.300 ^a,A^	10.20 ± 1.087 ^b,A^
		Yes (Group C)	11	10.51 ± 0.868 ^a,A^	9.14 ± 1.280 ^b,A^
	Surgery	No (Group B)	11	11.30 ± 0.819 ^a,B^	10.33 ± 0.850 ^b,B^
		Yes (Group D)	11	10.99 ± 0.591 ^a,B^	10.64 ± 0.793 ^b,B^

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
