# Peer review of "Impact of a Topical Anaesthesia Wound Management Formulation on Pain, Inflammation and Reduction of Secondary Infections after Tail Docking in Lambs"

_animals, 2020, doi:10.3390/ani10081255_

Round 1

Reviewer 1 Report

The topic of the paper is of big interest and the Authors addressed it presenting the results of experimental research performed on 44 lambs. The results may have an effective impact on ovine husbandry practices.

Just three questions regarding minor issues:

1)        How do the Authors explain the high prevalence of infection (82%) they observed after tail docking? Can the value they recorded to be considered “normal” for the ovine husbandry system in Spain?

2)        It is not clear if the values of SAA prior to the tail docking (S0) in “Traditional” groups are or not statistically different from “Surgery” groups ones.

3)        What is the cost of Tri-Solfen treatment?

Please, correct the following sentences:

row 35 <<with groups C and D groups immediately …>> edit to <<with groups C and D immediately …>>

row 81 <<The concentration of ….  change in response…>> edit to <<The concentration of ….  changes in response…>>

row 260 <<without general anaesthesia (groups A&D)..>> edit to <<without general anaesthesia (groups A&C)..>>

Author Response

Dear reviewer,

We would like to thank you for the work done. Here below you have the answers to your questions and suggestions:

  • How do the Authors explain the high prevalence of infection (82%) they observed after tail docking? Can the value they recorded to be considered “normal” for the ovine husbandry system in Spain? In Spain, the traditional method described in this article is generally used, but after the procedure spray antibiotics are used to reduce secondary infections. The high degree of infection found in the lambs in the study is probably due to the housing conditions of the animals and that no other product has been used to alleviate the infections.
  • It is not clear if the values of SAA prior to the tail docking (S0) in “Traditional” groups are or not statistically different from “Surgery” groups ones. Although they started with different concentrations of SAA, there were no significant differences between the groups.
  • What is the cost of Tri-Solfen treatment? The individual cost of Tri-Solfen is 0.5€. This will be better explained in the manuscript.

Please, correct the following sentences:

row 35 <<with groups C and D groups immediately …>> edit to <<with groups C and D immediately …>>. This has been done.

row 81 <<The concentration of ….  change in response…>> edit to <<The concentration of ….  changes in response…>> Corrected.

row 260 <<without general anaesthesia (groups A&D)..>> edit to <<without general anaesthesia (groups A&C)..>> Corrected

Thank you again and kind regards.

Reviewer 2 Report

This study compared several surgical approaches with and without general anaesthesia and with and without surgical stitching, plus examined the efficacy of applying a commercially available Tri-Solfen®. The paper is well written. I found just some minor points which I addressed below:

Line 21: Would it be not better to speak from different surgical approaches due to the fact that you are listening so many different groups you are comparing.

Line 33: Here we have now 4 groups. Please make line 21 and 33 equal.

Line 299: ... general to local ... (we don't see waht You compare)

Line 324: I go along with the interpretation of the authors but want to put another point to their discussion. The Swiss animal protection law does not allow any procedure on animals with pain. The discussion in Switzerland goes in the direction if it is sufficient to give the animal a post-op anesthesia or does it need a pre-op anesthesia to go according to the law?

Line 395:  … significant. This … (point is missing)

Line 427:  SAA? (not SSA)

Author Response

Dear reviewer,

We would like to thank you very much for the work done. Here below you can find the answers to your suggestions.

Line 21: Would it be not better to speak from different surgical approaches due to the fact that you are listening so many different groups you are comparing. Yes, you are right, this has been modified in the manuscript.

Line 33: Here we have now 4 groups. Please make line 21 and 33 equal. The word “two” has been replaced by “several”.

Line 299: ... general to local ... (we don't see waht You compare). Yes, we are comparing both, general and local anaesthesia and so it has been added to the text.

Line 324: I go along with the interpretation of the authors but want to put another point to their discussion. The Swiss animal protection law does not allow any procedure on animals with pain. The discussion in Switzerland goes in the direction if it is sufficient to give the animal a post-op anesthesia or does it need a pre-op anesthesia to go according to the law? This study suggests that post-op anaesthesia with TS would reduce pain considerably.

Line 395:  … significant. This … (point is missing). Thank you. This has been added.

Line 427:  SAA? (not SSA). Changed.

Thank you again and kind regards.